# The Effect of BEOL Design Factors on the Thermal Reliability of Flip-Chip Chip-Scale Packaging

**DOI:** 10.3390/mi16020121

**Published:** 2025-01-22

**Authors:** Dejian Li, Bofu Li, Shunfeng Han, Dameng Li, Baobin Yang, Baoliang Gong, Zhangzhang Zhang, Chang Yu, Pei Chen

**Affiliations:** 1Beijing Smart-Chip Microelectronics Technology Company Limited, Beijing 100089, China; 2School of Integrated Circuits, Tsinghua University, Beijing 100084, China; 3College of Mechanical & Energy Engineering, Beijing University of Technology, Beijing 100124, China; 4School of Mathematics, Statistics and Mechanics, Beijing University of Technology, Beijing 100124, China

**Keywords:** chip package interaction, back-end-of-line reliability, flip-chip chip-scale packaging, low-k cracking, thermal stress

## Abstract

With the development of high-density integrated chips, low-k dielectric materials are used in the back end of line (BEOL) to reduce signal delay. However, due to the application of fine-pitch packages with high-hardness copper pillars, BEOL is susceptible to chip package interaction (CPI), which leads to reliability issues such as the delamination of interlayer dielectric (ILD) layers. In order to improve package reliability, the effect of CPI at multi-scale needs to be explored in terms of package integration. In this paper, the stress of BEOL in the flip-chip chip-scale packaging (FCCSP) model during thermal cycling is investigated by using the finite-element-based sub-model approach. A three-dimensional (3D) multi-level finite element model is established based on the FCCSP. The wiring layers were treated by the equivalent homogenization method to ensure high prediction accuracy. The stress distribution of the BEOL around the critical bump was analyzed. The cracking risk of the interface layer of the BEOL was assessed by pre-cracking at a dangerous location. In addition, the effects of the epoxy molding compound (EMC) thickness, polyimide (PI) opening, and coefficient of thermal expansion (CTE) of the underfill on cracking were investigated. The simulation results show that the first principal stress of BEOL is higher at high-temperature moments than at low-temperature moments, and mainly concentrated near the PI opening. Compared with the oxide layer, the low-k layer has a higher risk of cracking. A smaller EMC thickness, lower CTE of the underfill, and larger PI opening help to reduce the risk of cracking in the BEOL.

## 1. Introduction

With the development of semiconductor processes, the interconnect delay in the back end of line (BEOL) increases significantly, resulting in a mismatch between the computational speed and the transmission bandwidth. In order to reduce the signal delay and power consumption of the chip, low-k dielectric materials are usually used in the BEOL layer of the chip to reduce the capacitive coupling effect [1,2]. However, due to the high porosity and weak mechanical properties of low-k materials, they are highly susceptible to chip package interaction (CPI), which, in turn, leads to reliability issues such as cracking and the delamination of the interlayer dielectric (ILD) layer in the BEOL [3,4,5,6,7]. Therefore, it becomes extremely important to ensure the integrity of the BEOL layer.

The Heterogeneous Integration Roadmap (HIR) Workshop [8] states that CPI occurs when the chip is subjected to factors such as thermal loading during the packaging process or during operation. Failures caused by CPI are mainly characterized by ILD delamination at the chip edge bumps, which is commonly seen as white spots in the scanning acoustic microscope (SAM) image that are, therefore, called “white bumps”. This phenomenon is mainly due to the inherent weakness of the mechanical properties of low-k materials and CPI, resulting in the BEOL layer fracture driving force exceeding the critical value of low-k. In addition, high-hardness copper pillars are used instead of solder balls in flip-chip chip-scale packaging (FCCSP) to enable fine-pitch chip packages, and this change also exacerbates the occurrence of BEOL delamination during the packaging process and reliability testing [9,10].

A number of experimental methods can provide insight into the location of deformation and cracking in the package. Thermal deformation in chip packages can be detected using the shadow moiré technique [11]. The phase difference between the interference fringes, combined with optical measurement techniques, enables the degree of deformation in the cross-sectional region of the chip to be calculated. Delamination near dangerous bumps inside the chip can be detected non-destructively using the SAM technique [12]. The detection accuracy of these experimental methods may be limited due to the fact that BEOL layers are usually on the nanometer scale. Finite element analysis (FEA) is an important tool for stress analysis, especially when dealing with multi-scale challenges. Therefore, a more comprehensive and accurate assessment of BEOL layer stress and cracking risk can be achieved through a combined experimental and simulation approach.

In order to solve the multiscale problem of FEA, researchers have commonly used multi-level sub-model approach to analyze the stress and cracking of the BEOL under CPI. Raghavan et al. [13] built a 2D flip-chip model to investigate the stress distribution and cracking risk of the BEOL layer during reflow process. Wang et al. [14,15] built a 3D fine sub-model of the BEOL layer to study the stress distribution of a 40 nm chip during the reflow process. It was found that the stresses of different interlayer metal layers (IMDs) fluctuated in a small range under the critical bump, while the first-principal stress of ILD layers increased gradually with the increase in the number of layers. Meanwhile, Zhang et al. [16] investigated the BEOL delamination of 45 nm chips during reflow from the perspective of fracture mechanics and discussed the effects of different low-k materials, solder, and package materials on the energy release rate (ERR). In addition, Chu et al. [17,18] compared the cracking risk of BEOL layers with different dielectric materials using mixed-signal chips. It was found that the interconnect layer of ELK (extremely low-k) material had the highest risk of cracking, and the effects of the wiring density and copper pillar structure of each interconnect layer of the BEOL were also analyzed. However, all of the above studies mainly focus on the analysis of stress and cracking of the BEOL layer during reflow process, and there are relatively few studies on the reliability process of thermal cycling.

Previous experiments have shown [19] that interfacial delamination was often observed in the BEOL layer of flip-chip ball grid array (FCBGA) packages under thermal cycling conditions. Liu et al. [20] investigated the risk of BEOL cracking during thermal cycling in 2D chips and found that the difference in the modulus and coefficient of thermal expansion (CTE) of the underfill significantly affects the ERR. However, the 2D model does not fully characterize the stress and delamination risk of the BEOL during thermal cycling.

In this paper, a multi-level sub-model approach is used to investigate the stress and cracking risk of the BEOL layer in 3D FCCSP packages during thermal cycling. Since the substrate and BEOL layers in FCCSP are composed of multiple types of materials and have a multi-scale structure and complex geometrical features of the copper wiring layer, to address these complexities, improve the computational and modeling efficiency, and meet the requirement of high prediction accuracy, the wiring layer is modeled with equivalent homogenization to determine the macroscopic response of the structure. The finite element model was validated by measuring the warpage of the FCCSP package using the shadow moiré method experimentally. The BEOL stress distribution around the critical bump was analyzed, and the ERR at the crack tip was calculated by inserting a pre-crack in the interface layer. The effects of the EMC thickness, PI opening, and CTE of the underfill on the ERR are further discussed.

## 2. Methods

### 2.1. Details of the Structure of the FCCSP Package

A FCCSP package with Cu pillars consisting of a 28 nm node chips was used as the finite element model. Figure 1a shows the SEM cross-section of the FCCSP package, which mainly consists of the silicon die, Cu pillar bumps, and cored substrate. The dimensions of the FCCSP are 15 mm × 15 mm × 0.86 mm, where the thickness of the EMC is 0.45 mm, the dimensions of the die are 5.48 mm × 5.84 mm × 0.15 mm, and the thickness of the substrate is 0.41 mm. The substrate used 13 layers of embedded trace substrate (ETS), mainly including the core layer, two solder mask (SM) layers, six prepreg (PP) dielectric layers, and six metal layers, which are filled with PP material. Figure 1b shows a detailed cross-section image of the Cu pillar bump structure, where the height of the Cu pillar is 30 μm, the thickness of the Ni layer is 3 μm, the height of the reflowed solder is 10 μm, and the bump shape is an elliptical cylinder with the dimensions of 45 μm × 60 μm, in addition to the thickness of the 5 μm PI layer. Figure 1c shows a cross-section of the BEOL structure with eight metal layers, metal layer 1 (M1) to metal layer 8 (M8), defined from top to bottom. The different metal layers are connected to each other through vias and filled with dielectric material. The first six layers are low-k layers (dielectric material is low-k material) and the last two layers are oxide layers (dielectric material is SiO_2_ material). Table 1 gives the details of the Cu pillar bump-type FCCSP package.

### 2.2. Multi-Level Sub-Model and Material Constants

The feature size of the FCCSP package is about 10^−2^ m and the BEOL is about 10^−9^ m. Considering that there is a seven-order-of-magnitude difference in the length scale, a multi-level sub-model approach is used in this study. There are four levels of the model, which are package level, bump level, BEOL level, and wafer level, as shown in Figure 2.

It is worth noting that the copper wiring layers have a very complex geometry that makes the simulation unfeasible. In order to simplify the finite element model, the metal layers of the substrate and the BEOL layers of the chip are modeled with equivalent homogenization, and the effective orthogonal anisotropic material properties of the metal layers are determined using rule of mixture (ROM) analysis. The percentage of metal in each layer was determined from the metal layout. The equations are as follows [21,22]:(1)$$E_{x}=E_{y}=\frac{E_{d}E_{m}}{E_{d}c_{m}+E_{m}c_{d}}$$(2)$$E_{z}=E_{d}c_{d}+E_{m}c_{m}$$(3)$$\nu_{xy}=\nu_{m}c_{m}+\nu_{d}c_{d}\left[\frac{1+\nu_{d}-\nu_{xz}\left(E_{d}/E_{z}\right)}{1-\nu_{d}^{2}+\nu_{d}\nu_{xz}\left(E_{d}/E_{z}\right)}\right]$$(4)$$\nu_{xz}=\nu_{yz}=\frac{E_{x}\left(c_{d}\nu_{d}+c_{m}\nu_{m}\right)}{E_{z}}$$(5)$$G_{xy}=\frac{E_{x}}{2\left(1+\nu_{xy}\right)}$$(6)$$G_{xz}=G_{yz}=\frac{G_{d}G_{m}}{G_{d}c_{m}+G_{m}c_{d}}$$(7)$$\alpha_{x}=\alpha_{y}=\alpha_{d}c_{d}\left(1+\nu_{d}\right)+\alpha_{m}c_{m}\left(1+\nu_{m}\right)-\alpha_{z}\nu_{xz}$$(8)$$\alpha_{z}=\frac{\alpha_{d}E_{d}c_{d}+\alpha_{m}E_{m}c_{m}}{E_{z}}$$
where *E* is Young’s modulus, *ν* is Poisson’s ratio, *G* is shear modulus, and *α* is CTE. The subscript *m* is copper and *d* is PP material or dielectric material. *c* is volume fraction. The subscripts “*x*”, “*y*”, and “*z*” are for X, Y, and Z directions, “*xy*” is in plane, and “*xz*” and “*yz*” are out of plane.

The package-level model was built using the commercial finite element software ABAQUS v2021 to analyze the thermal deformation of the whole FCCSP package. The model includes EMC, silicon die, underfill, Cu pillar, solder, and substrate layers, as shown in Figure 2a. The details of the BEOL layer are not considered because its thickness is too small compared to the whole package. Based on the simulation results of the package-level model, critical bump locations were identified. The bump-level model focuses on a bump region with more structural details compared to the package-level model. The main structures include PI layer, NI layer, passivation layer, aluminum pads, and BEOL equivalent layer, as shown in Figure 2d. This level model considers the BEOL layer but does not consider the detailed structures in the BEOL layer. The effective parameters of the BEOL equivalent layer were calculated based on the method of the literature [23]. The BEOL-level model focuses on the region between the die and the bump (a small region of the bump-level model). It includes the BEOL layer, the partial silicon die, the partial aluminum pad, and the passivation layer, as shown in Figure 2e. At this level, the BEOL layer includes the equivalent IMD and ILD layers without the detailed interconnect structure, where the first six layers are low-k layers and the last two layers are oxide layers. The detailed wafer-level model is further scaled up from the BEOL level to a sub-model that focuses on the BEOL-level model region with the highest peeling stress. Based on the interconnect layer structure in the SEM image in Figure 1c, a simplified wafer-level model was built, as shown in Figure 2g. The boundary conditions for each level of the sub-model were obtained by the displacements obtained from the upper-level model. The material parameters of the FCCSP package are given in Table 2, where all materials are considered as linear elastic materials.

### 2.3. FE Mesh and Boundary Load

Low-k materials are usually brittle materials with weak tensile strength. Large tensile stresses may lead to fracture of low-k materials, which can easily propagate through the interface between the ILD layer and IMD layer [24]. Therefore, the cracks are prefabricated at the interface for fracture analysis. The meshing of the wafer-level model is shown in Figure 3a, and the element type used is the eight-node linear hexahedral element (C3D8R). In order to ensure numerical accuracy, a dense mesh is set up at the location near the crack tip, as shown in Figure 3b. To capture the stress singularity, 20-node secondary hexahedral elements (C3D20R) were used near the crack tip. Mesh convergence studies were performed on the model by changing the mesh density. The energy release rate is used as the convergence parameter for different mesh densities.

The J-integral in ABAQUS v2021 can be extended to represent the pointwise energy release rate along the crack front as [25]:(9)$$J\left(s\right)=\underset{\Gamma \to 0}{\lim}\int_{\Gamma }n\cdot H\cdot qd\Gamma$$

For a virtual crack advance *λ*(*s*) of a 3D crack, the energy release rate is given by(10)$$\bar{J}=\int_{L}J\left(s\right)\lambda \left(s\right)ds=\underset{\Gamma \to 0}{\lim}\int_{A}\lambda \left(s\right)n\cdot H\cdot qdA$$
where Γ is any contour from the bottom crack surface around the crack tip to the top surface. *L* is the crack front. *dA* is a surface element on a vanishingly small tubular surface enclosing the crack tip (*dA* = *dsdΓ*), and *n* is the outward normal to *dA*. *H* can be expressed as(11)$$H=W\delta -\sigma \frac{\partial u}{\partial x}$$
where *σ* is the stress tensor, *u* is the displacement vector, and *δ* is the Kronecker symbol. For elastic material behavior, *W* is elastic strain energy.

In this paper, the ERR of the crack front is extracted directly from ABAQUS. An average of 10 contours is used to calculate the ERR. The ERR obtained from the first two contours is neglected to improve the accuracy. The thermal cycling process from 125 °C to −55 °C is simulated and the temperature profile is shown in Figure 4. In order to prevent the rigid body from moving, a three-point displacement constraint method is used to constrain the six degrees of freedom of the model. The center point at the bottom of the FCCSP constrains the displacements in the X, Y, and Z directions, the center point at the top of the FCCSP constrains the displacements in the X and Y directions, and the center point at the edge of the FCCSP in the X direction constrains the displacements in the Y direction.

## 3. Results and Discussion

### 3.1. FCCSP Package Warpage Verification

Warpage values at the four corner point locations of the FCCSP package were first measured by shadow moiré experiments at temperatures ranging from 30 °C to 260 °C. The experimental setup is as follows: First, the grating cover inside the device is pulled up to a vertical position, and then the sample is placed on the carrier table, making sure that the bottom side of the substrate is facing up. The bottom of the carrier stage is equipped with a heating tube for uniform heating of the sample. At the same time, a thermocouple is attached to the sample surface to measure its temperature in real time. After the sample was placed, the grating sheet cover was closed and a reference grating pattern was projected onto the sample surface. Next, a charge-coupled device (CCD) camera was used to capture the reflected light and analyze the moiré fringes formed by the interference to infer the warpage of the sample. In the high-temperature stage, some material parameters will change significantly, resulting in a significant change in the warpage tendency, so focus on observing the change of warpage at high temperature. Since the warpage values of the four corners of the package extracted in the simulation were almost the same, only one of the corner points was taken as the measurement point for the simulation. The simulated warpage values were compared with the experimental values, as shown in Figure 5. Warpage tends to decrease in the early stages of warming, mainly due to the fact that the CTE of the EMC is less than the CTE of the substrate before *T*_g_ of the EMC (12 ppm/°C for the EMC, 17 ppm/°C for the copper in the substrate, 14 ppm/°C for the PP, and slightly less than the CTE of the EMC for the core at 10.5 ppm/°C, but overall, the CTE of the substrate is greater than the CTE of the EMC). As a result, the substrate on the upper surface expands more than the EMC, resulting in a decreasing warpage trend. In the later stages of warming, the CTE of the EMC increases from 12 ppm/°C to 45 ppm/°C, which exceeds the CTE of the substrate, so the substrate expands less than the lower EMC, resulting in a gradual upward trend in warpage. It can be seen that the simulated values were close to the experimental values, validating the multi-level FEA model.

### 3.2. Locating the Critical Bump

Since the thermal stress is proportional to the distance from the center of the chip, the outermost bumps are critical. It can also be seen in Figure 6a that the closer to the position of the bumps at the four corner points, the greater the deformation. However, as shown in Figure 6b, the location distribution of the bumps is not regular. In order to find the vulnerable locations of the BEOL layer, bump-level models were placed at each of the four corners of the chip, along the horizontal centerline, along the vertical centerline, and near the center of the bumps, with each corner containing two bumps. The side length of this bump-level model is 120 μm to ensure that the bumps can be completely covered. Displacement interpolation is applied to the boundaries of the bump-level sub-model.

The maximum first-principal stresses in the BEOL equivalent layer were extracted as shown in Figure 7a. It is found that the stresses located at the corners of the chip are higher than those located at the horizontal centerline, vertical centerline, and center position, where the BEOL at the center point is subjected to the least stress. For the eight locations at the corners, it is found that the maximum first-principal stresses at location 6 and 8 are relatively high, with a stress of about 445 MPa, while the stresses at other positions average around 430 MPa, with a difference of about 15 MPa. It can be found that the bump placement angle has an effect on the stress of the BEOL. By changing the bump placement angle at location 8 and keeping the other bumps unchanged, the stress distribution of the BEOL at different placement angles was calculated, as shown in Figure 7b. It can be seen that the stresses are relatively high when the bump placement angle is horizontal or vertical. The stresses are relatively low when the bump placement angle is from 30° to 60°. Circular bumps with diameters of 45 μm and 60 μm were also calculated. It can be seen that for the 45 μm diameter bump, the maximum stress is 455 MPa, which is higher than that of the elliptical bump. For the 60 μm diameter bump, the maximum stress is 436 MPa, which is in between the stresses of the elliptical bumps at different angles. The BEOL layer of the bump at location 8 has the highest stress and is identified as the critical bump position.

### 3.3. Stress Distribution of BEOL Layers

The BEOL-level model was positioned at the location of the BEOL equivalent layer of the bump-level model, as shown in Figure 2e. The edge length of the model is 100 μm, and the center is aligned with the center of the bump. Due to the fact that the ILD is a via layer with a low percentage of copper, each ILD layer may constitute a crack extension path, so the focus is on analyzing the stress distribution of the ILD equivalent layer. The stress-free temperature for all materials is assumed to be 25 °C. The maximum principal stress contour of the ILD equivalent layer at the 125 °C and −55 °C moments are shown in Figure 8. It can be seen that at 125 °C, the stress is concentrated around the left of center region along the 45° diagonal. However, at −55 °C, the stress is relatively minimal in the center region.

In order to characterize the stress distribution in the different ILD equivalent layers, a 45° diagonal path was defined in each ILD equivalent layer. Figure 9 shows the stress distribution along the diagonal at the 125 °C and −55 °C moments. At 125 °C, the stresses in both the low-k layer and the oxide layer reach a maximum at −12 μm from the center of the bump (near the edge of the PI opening), with the stress in the oxide layer being relatively high at about 340 MPa. At −55 °C, the stresses in both the low-k layer and the oxide layer reach a minimum at −9 μm from the center of the bump, where the stress in the oxide layer is relatively low.

The location of the wafer-level model is positioned at the location of the maximum principal stress of the BEOL-level model. The maximum first-principal strain contour of the dielectric material in the ILD layers at 125 °C is shown in Figure 10a. It can be seen that the strain in the low-k layers is larger than that in the oxide layers because the modulus of elasticity of the low-k layers in the BEOL is lower than that of the oxide layers. Figure 10b shows the distribution of the principal strain in the ILD layers, and it can be seen that the strain at 125 °C is significantly higher than that at −55 °C, indicating a higher risk of cracking at high temperatures.

### 3.4. ERR of Pre-Cracks in the Different Layers

A 2D horizontal crack with a length of 1.3 μm and a width of 1.0 μm was inserted in the interface between the IMD and ILD, as mentioned before in Figure 3. The cracks in the different layers have the same dimensions. The maximum ERR of the crack tip at different interface layers is shown in Figure 11a. It can be seen that the ERR at 125 °C is higher than that at −55 °C, indicating a higher risk of cracking at high-temperature moments. At 125 °C, the ERR increases first with the number of layers and then decreases, with the highest ERR in the sixth layer. The cracking risk of each layer can also be analyzed in terms of strain energy density. The average strain energy densities of the different layers are shown in Figure 11b. It can be seen that the trend of the strain energy density in the ILD layers is consistent with the trend of the ERR. At 125 °C, the average strain energy density is generally higher than the value at −55 °C. Meanwhile, the strain energy density of the low-k layers is higher compared to that of the oxide layers. Considering that the critical ERR of low-k material is 6 J/m^2^ [26] and the critical ERR of oxide material is 14 J/m^2^ [27], it is evident that the sixth layer has the highest critical ERR, which indicates that the sixth layer is most prone to fracture.

### 3.5. Influence of Structural Dimension and Material Parameters on ERR

Optimizing the package structure and material parameters to reduce the stress in the BEOL layer contributes to the reliability of the FCCSP package. In the parametric study, three important parameters were selected: the EMC thickness, the CTE of the underfill, and the PI opening. The EMC thickness and the CTE of the underfill were modified starting from the package-level model. The EMC thickness was 400 μm, 450 μm, and 500 μm, respectively. The underfill was modified to the CTE after *T*_g_, which was 75 T, 95 ppm/°C, and 115 ppm/°C. The PI openings were modified starting in the bump-level model with diameter dimensions of 24 μm, 30 μm, and 36 μm, respectively. The wafer-level models are positioned at the location of the maximum principal stresses of the BEOL-level models, and the ERR of the sixth ILD layer is chosen as the standard of comparison. As can be seen in Figure 12a, the ERR of the 400 μm thick EMC decreases by 7.2% and the ERR of the 500 μm thick EMC increases by 6.0% compared to the ERR of the 450 μm thick EMC. This can be attributed to the effect of the thickness of the EMC on the warpage of the package-level model, with the bumps at the corner locations being more significantly affected by the warpage. The bumps are subjected to greater shear loads and more stress is transferred to the BEOL layer. As can be seen in Figure 12b, the ERR decreases by 20.1% for the low CTE and increases by 22.3% for the high CTE compared to the CTE of 95 ppm/°C for the underfill. This situation can be attributed to the high CTE of the underfill when the FCCSP package experiences high temperatures, which exacerbates the degree of CTE mismatch between the bumps, leading to larger tensile stresses at the BEOL layer. As can be seen in Figure 12c, the ERR increases by 13.7% for the 24 μm diameter and decreases by 5.4% for the 36 μm diameter compared to the PI opening with a diameter of 30 μm. This situation can be attributed to the fact that the underfill expansion produces tensile stresses near the bump, which are mainly concentrated within the PI opening region. A larger PI opening reduces the concentration of tensile stresses.

## 4. Conclusions

In this paper, the reliability of the BEOL layer of FCCSP packages during thermal cycling is studied through a multi-level sub-model approach. A multiscale and multi-level finite element model in 3D is built for CPI analysis, concentrating at BEOL layer stress analysis and cracking risk assessment. The results of CPI analysis are as follows,

(1)The BEOL layer located at the chip corner locations has the highest stress compared to the center location, horizontal centerline, and vertical centerline locations. In addition, the BEOL layer is subjected to relatively high stresses when the bumps at the chip corner locations are placed in the vertical or horizontal direction.(2)At the critical bump location, the first-principal stresses in the BEOL layer are larger at high-temperature moments compared to low-temperature moments and are mainly concentrated near the PI opening.(3)When the defects are located in the high-principal-stress region of the BEOL layer, the cracking risk of the low-k layers is significantly higher than that of the oxide layers. In particular, the risk of cracking is highest in the layer connecting the low-k layers to the oxide layers.(4)A smaller EMC thickness and underfill with a low CTE can reduce the cracking risk of the BEOL layer. Meanwhile, a larger PI opening helps to alleviate the stress concentration in the BEOL layer.

In conclusion, the reliability of the BEOL layer of FCCSP packages during thermal cycling has been investigated by numerical methods. By combining these findings with experimental test methods, as referenced in [28,29], the cracking risk of the BEOL layer may be evaluated more accurately. This integrated approach can provide valuable recommendations for optimizing design in line with established testing standards.

## Figures and Tables

**Figure 1 micromachines-16-00121-f001:**
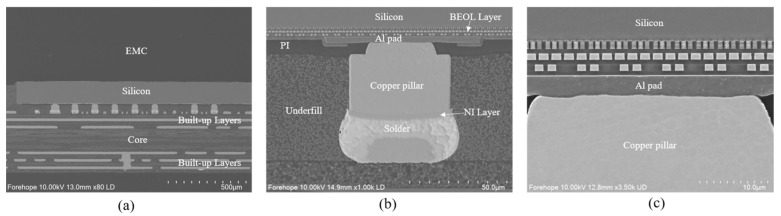
SEM images of FCCSP package: (**a**) cross-section of FCCSP; (**b**) zoom-in cross-section of bump location; (**c**) detailed cross-section of BEOL layer on the bump.

**Figure 2 micromachines-16-00121-f002:**
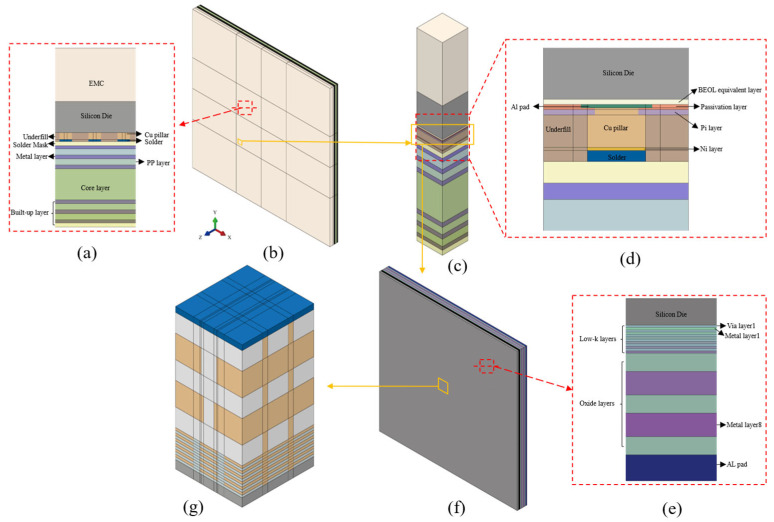
FEA model with multi-level sub-model approach: (**a**) cross-section of the package-level model; (**b**) package-level model; (**c**) bump-level model; (**d**) cross-section of the bump-level model; (**e**) cross-section of the BEOL-level model; (**f**) BEOL-level model; (**g**) wafer-level model.

**Figure 3 micromachines-16-00121-f003:**
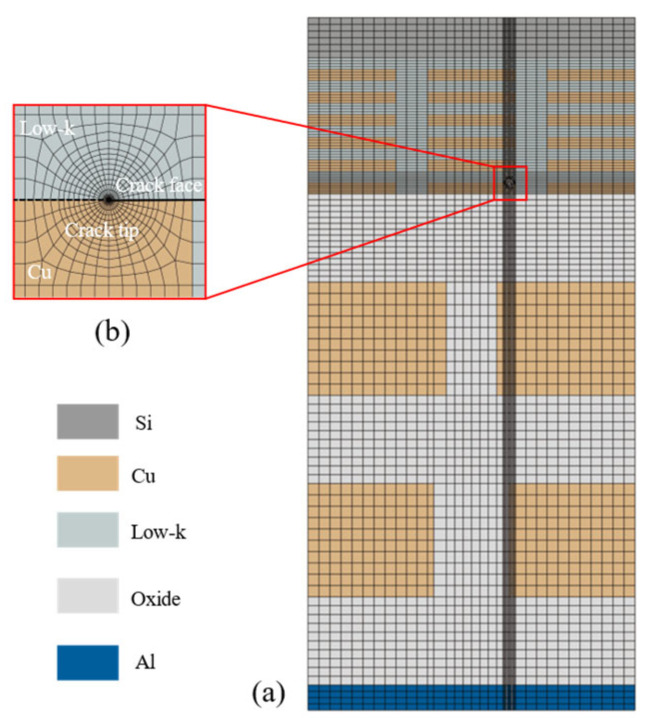
Schematic diagram of finite element meshes: (**a**) meshes of the wafer-level model; (**b**) meshes of crack tip.

**Figure 4 micromachines-16-00121-f004:**
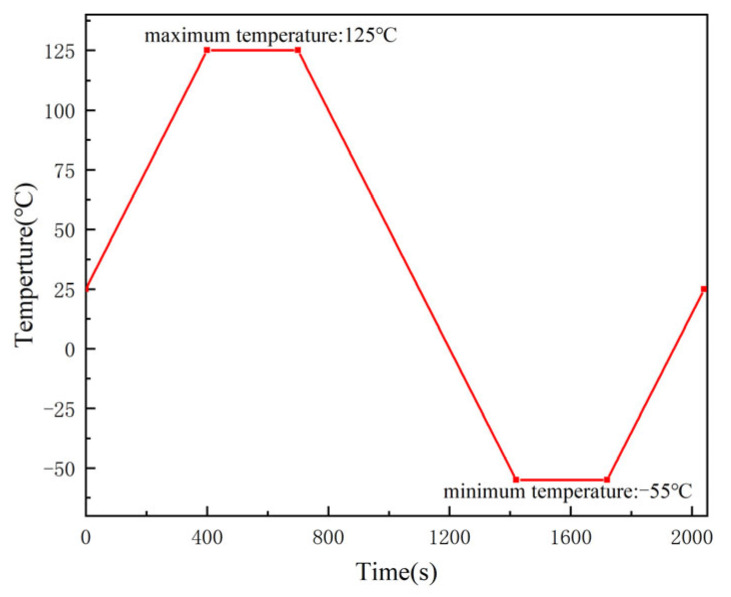
Thermal cyclic loading curve.

**Figure 5 micromachines-16-00121-f005:**
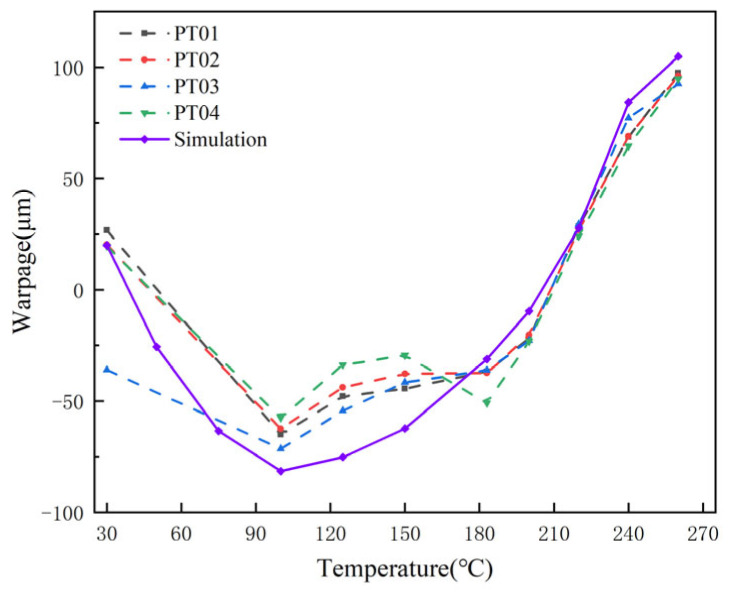
Comparison of FEA and shadow moiré results of thermal deformation for the FCCSP package.

**Figure 6 micromachines-16-00121-f006:**
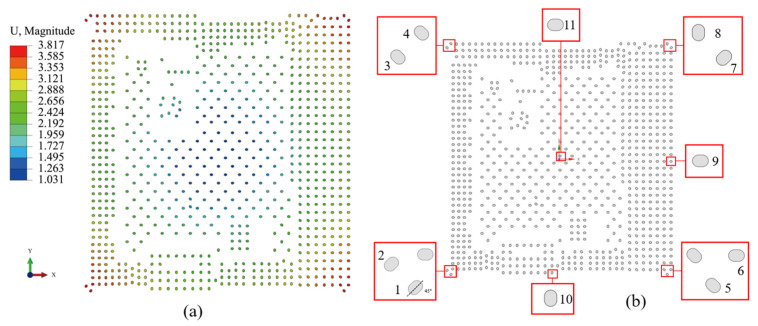
(**a**) Bumps deformation contour; (**b**) schematic diagram of bumps location for FCCSP package.

**Figure 7 micromachines-16-00121-f007:**
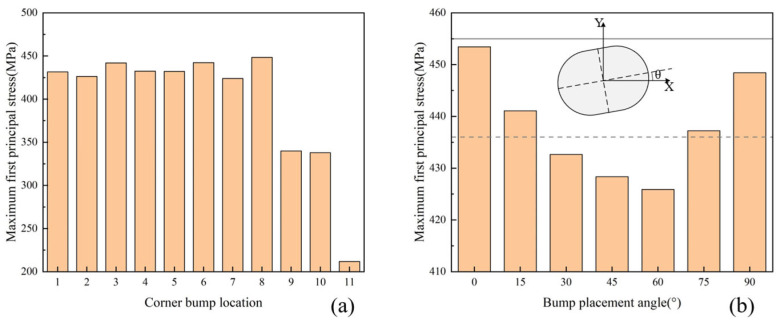
(**a**) Maximum first-principal stresses in BEOL equivalent layers at different bump locations; (**b**) stress distribution of bump at different placement angles (the dashed line is the bump with a diameter of 60 μm and the solid line is the bump with a diameter of 45 μm).

**Figure 8 micromachines-16-00121-f008:**
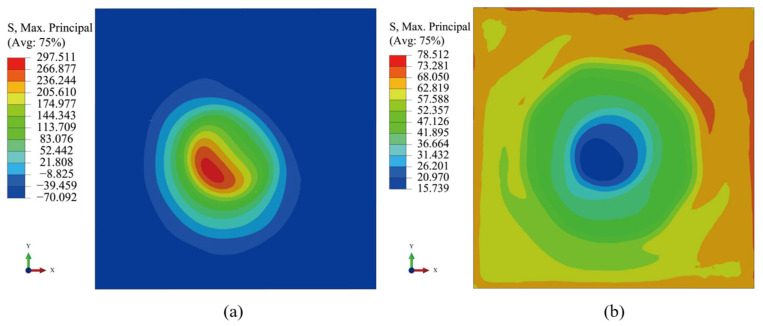
Maximum principal stress contour of the ILD equivalent layers: (**a**) 125 °C; (**b**) −55 °C.

**Figure 9 micromachines-16-00121-f009:**
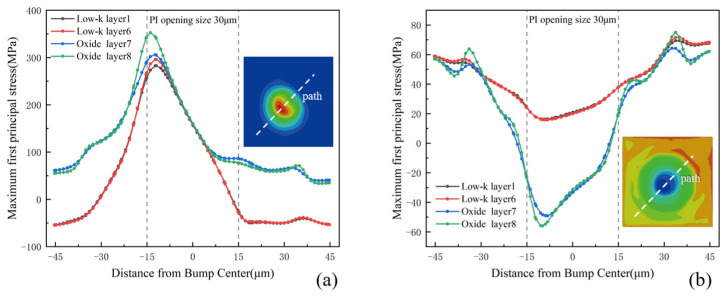
Stress distribution in the ILD equivalent layers along the diagonal path: (**a**) 125 °C; (**b**) −55 °C.

**Figure 10 micromachines-16-00121-f010:**
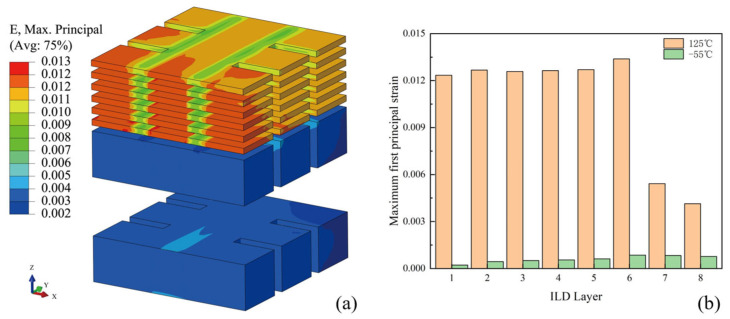
(**a**) Maximum first-principal strain contour of ILD layers at 125 °C; (**b**) maximum first-principal strain in different ILD layers.

**Figure 11 micromachines-16-00121-f011:**
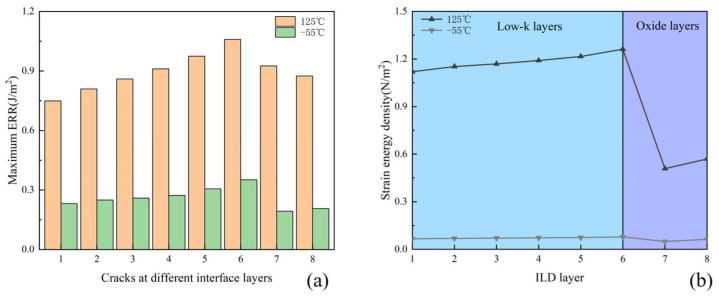
(**a**) Maximum ERR at the crack front for different interfacial layers; (**b**) average strain energy density of the ILD layers.

**Figure 12 micromachines-16-00121-f012:**
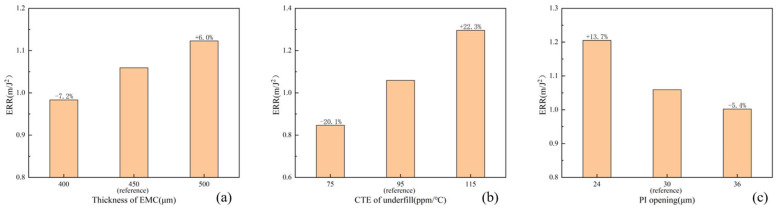
Effects of material and geometric parameters: (**a**) thickness of EMC; (**b**) CTE of underfill; (**c**) PI opening.

**Table 1 micromachines-16-00121-t001:** Dimensions of the Cu pillar bump-type FCCSP package.

Item	Dimension
EMC size (mm^3^)	15 × 15 × 0.45
Die size (mm^3^)	5.48 × 5.84 × 0.15
Substrate size (mm^3^)	15 × 15 × 0.41
Substrate core thickness (μm)	150
Substrate layer number	13
Bump height (μm)	58 (before flip-chip), 43 (after flip-chip)
PI Layer thickness (μm)	5
PI opening (μm)	30
Al pad thickness (μm)	3
BEOL Layer thickness (μm)	5

**Table 2 micromachines-16-00121-t002:** Material properties of FCCSP package.

Material	Young’s Modulus (GPa)	Poisson’s Ratio	CTE (ppm/°C)	*T*_g_ (°C)
Si	131	0.3	2.8	–
Cu	120.66	0.345	17	–
Al	71	0.33	23	–
Ni	207	0.3	13	–
Pi	3.5	0.34	35	–
Passivation	210	0.27	2	–
SiO_2_	80	0.17	0.68	–
Low-k	25	0.3	18	–
Solder	88.53–0.142 *T*	0.3	22	–
Solder mask	6.2 (*T* < *T*_g_) 0.23 (*T* > *T*_g_)	0.29	60 (*T* < *T*_g_) 130 (*T* > *T*_g_)	114
EMC	20 (*T* < *T*_g_) 1.1 (*T* > *T*_g_)	0.26	12 (*T* < *T*_g_) 45 (*T* > *T*_g_)	150
Underfill	7 (*T* < *T*_g_) 0.2 (*T* > *T*_g_)	0.3	32 (*T* < *T*_g_) 95 (*T* > *T*_g_)	110
PP	13 (*T* < *T*_g_) 7.4 (*T* > *T*_g_)	0.23	14 (*T* < *T*_g_) 6 (*T* > *T*_g_)	230
Core	25 (*T* < *T*_g_) 14 (*T* > *T*_g_)	0.2	10.5 (*T* < *T*_g_) 4 (*T* > *T*_g_)	230

## Data Availability

The original contributions presented in this study are included in the article. Further inquiries can be directed to the corresponding author.

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
