# Peer review of "The Effect of BEOL Design Factors on the Thermal Reliability of Flip-Chip Chip-Scale Packaging"

_micromachines, 2025, doi:10.3390/mi16020121_

Round 1

Reviewer 1 Report

Comments and Suggestions for Authors

Authors numerically investigate stresses in chip BEOL when subject to flip-chip assembly with underfill application. Authors built a FEM 3D model focusing on area with grown Cu pillar, considering variations of polyimide opening size, BEOL dielectric stack, mold compound thickness and underfill CTE. Given model was used to investigate chip BEOL cracking at thermal cycling reliability test. The study brings interesting conclusions, I recommend it for publication after few minor comments are addressed.

1. Page 2, line 54. Authors claim "There is great uncertainty and difficulty in measuring the stress and cracking location of the BEOL layer using experimental methods. Therefore, Finite Element Analysis (FEA) is usually used for correlation analysis." I don't fully agree with this statement, a die warpage analysis before flip-chip and SAM imaging after flip-chip, that authors already mentioned before, can aid in failure analysis of cracking. Die warpage profiles from mechanical profiler scan or white-light interferoimetry will show absolute bow and macro stress vectors in the chip, and SAM will pinpoint failing bumps. Saying that, I agree that FEM modeling is a good tool for stress prediction, and it is best to calibrate model with observations from die warpage and SAM inspections.

2. Page 3, line 105. Authors mention bumps have elliptical shape of 45x60 um, which seems unusual from typical round shape bumps. Can authors ellaborate a bit why elliptical shape was used here?

3. Page 3, table 1. Authors mention bump height of 58 um before reflow and 43 um after reflow. It is worth mentioning that after reflow here means after flip-chip is done, as normally first round of reflow is done after Cu pillars are grown, before flip-chip, to smoothen top Sn layer. In this instance, bump height typically slightly increases, depending on Cu pillar aspect ratio.

4. Page 3, line 122. Authors mention BEOL stack is too complex to keep exact features to simulate together with package, due to size mismatch, and propose to homogenize BEOL mechanical properties using known Cu density for each metal layer. However, it is not clear if whole 8-layer BEOL stack is homogenized, or homogenization takes place for each layer separately. I mention this here because each metal layer can be quite different in thickness, and it is worth knowing which ILD layers are most sensitive to stresses.

5. Page 8, line 214. Authors highlight that elliptical bump placement angle has a significant impact on bump stress, from ~453 MPa at 0 deg down to ~426 MPa at 60 deg. That would be interesting to benchmark against round bumps of 45 and 60 um diameters.

6. Page 9, line 234. Authors show that stresses in both the low-k layer and the oxide layer at -15 μm from the center of the bump achieve max value, any explanation why not at center point?

General comments:

1. In chapter 3.5 authors check how variations in EMC thickness, mold compound CTE and polyimide opening size affect bump and BEOL stress. Interesting findings, however, I was bit surprised authors had so small PI openings comparing to bump size. It would be even more interesting to see bump and BEOL stresses when different loading of Cu layers is used in BEOL. Normally one is quite limited in design freedom here, but checking with min and max Cu densities allowed by foundry design rules would be interesting.

2. I suggest authors to enlarge legends for figures 6, 8, 9 ,10; they are difficult to look at.

Author Response

All of the modifications corresponding to current reviewer are in blue color in the manuscript.

Comments 1: Page 2, line 54. Authors claim "There is great uncertainty and difficulty in measuring the stress and cracking location of the BEOL layer using experimental methods. Therefore, Finite Element Analysis (FEA) is usually used for correlation analysis." I don't fully agree with this statement, a die warpage analysis before flip-chip and SAM imaging after flip-chip, that authors already mentioned before, can aid in failure analysis of cracking. Die warpage profiles from mechanical profiler scan or white-light interferoimetry will show absolute bow and macro stress vectors in the chip, and SAM will pinpoint failing bumps. Saying that, I agree that FEM modeling is a good tool for stress prediction, and it is best to calibrate model with observations from die warpage and SAM inspections.

Response 1: Thank you for your valuable feedback. We appreciate your insights regarding the utility of die warpage analysis and SAM imaging in failure analysis. We acknowledge that these experimental techniques can provide useful information for understanding stress and cracking in the BEOL layer. In response to your suggestion, we have clarified our statement to emphasize the complementary role of Finite Element Analysis (FEA) alongside these experimental methods, particularly in regards to the multiscale challenges present in analyzing BEOL layers. And revise the corresponding content in the third paragraph of the manuscript.

Comments 2: Page 3, line 105. Authors mention bumps have elliptical shape of 45x60 um, which seems unusual from typical round shape bumps. Can authors elaborate a bit why elliptical shape was used here?

Response 2: Thank you for your thoughtful question regarding the elliptical shape of the bumps (45x60 μm) mentioned in our manuscript. We appreciate the opportunity to elaborate on this design choice.

Generally, the bumps are round with different sizes. The micro-bumps with diameters of 70 μm or 80 μm are commonly used as “larger bump”, which are used for thick substrates with more than eight layers.

However, challenges with Larger Bumps: In the context of FCCSP packaging, reaching more than six substrate layers is challenging in FCCSP packaging. As a result, the typical bump size tends to be around 45 μm, which could cut the cost. But 45 μm bumps can pose reliability issues, such as stress concentration leading to bump cracking.

Considering cost and reliability, we opted for elliptical bumps measuring 45x60 μm. This design choice allows us to increasing the bump size slightly in the purpose of improving the reliability, but still control the cost The good stress reliability of elliptical bump is also confirmed by simulation in the response of comments 5.

Comments 3: Page 3, table 1. Authors mention bump height of 58 um before reflow and 43 um after reflow. It is worth mentioning that after reflow here means after flip-chip is done, as normally first round of reflow is done after Cu pillars are grown, before flip-chip, to smoothen top Sn layer. In this instance, bump height typically slightly increases, depending on Cu pillar aspect ratio. 3

Response 3: Thank you for your insightful comments. Actually, the bump height of 58 μm is already after the first round reflow and Cu pillar has already grown, which has be changed to “before flip-chip” in the text, and the bump height of 43 μm refers to the height “after flip-chip”.

Comments 4: Page 3, line 122. Authors mention BEOL stack is too complex to keep exact features to simulate together with package, due to size mismatch, and propose to homogenize BEOL mechanical properties using known Cu density for each metal layer. However, it is not clear if whole 8-layer BEOL stack is homogenized, or homogenization takes place for each layer separately. I mention this here because each metal layer can be quite different in thickness, and it is worth knowing which ILD layers are most sensitive to stresses.

Response 4: Thank you for your constructive feedback regarding the homogenization of the BEOL stack. We appreciate your insight into the complexities of the different layers and the necessity for clarity in our methodology. To clarify, we do not homogenize the whole 8-layer BEOL stack simply, and we apply equations 1-8 to homogenize the properties for EACH individual layer of the BEOL stack, yielding equivalent parameters that are then used in the BEOL submodel. Following this, we utilize the equations referenced in literature 20 to obtain a homogenized equivalent parameter for the entire 8-layer BEOL stack, which is then applied to the bump-level submodel.

Comments 5: Page 8, line 214. Authors highlight that elliptical bump placement angle has a significant impact on bump stress, from ~453 MPa at 0 deg down to ~426 MPa at 60 deg. That would be interesting to benchmark against round bumps of 45 and 60 um diameters.

Response 5: Thank you for your valuable suggestion to benchmark the stress results of elliptical bumps against round bumps with diameters of 45 μm and 60 μm.

In response to your recommendation, we have modeled the round bumps with diameters of 45 μm and 60 μm. The computed maximum principal stresses in the BEOL layer for these bump diameters are as follows.

For 45 round bumps of 45 μm diameter, the maximum stress is 455 MPa, which is higher than that of the elliptical bumps. For round bumps of 60 μm diameter, the maximum stress is 436 MPa, which lies between the stresses of the elliptical bumps at different angles.

Furthermore, we have updated Figure 7b in the manuscript to include benchmark lines for the round bumps. The dashed line represents the stress for the round bumps of 60 μm diameter, while the solid line corresponds to the round bumps of 45 μm diameter. This enhancement provides a clearer comparison of the stress distribution between elliptical and round bump geometries.

Comments 6: Page 9, line 234. Authors show that stresses in both the low-k layer and the oxide layer at -15 μm from the center of the bump achieve max value. Any explanation why not at center point?

Response 6: Thank you for your insightful comment.

In our original analysis, we reported stress values extracted at a distance of -15 μm from the center. However, upon re-evaluating the data and extracting measurements from more data points of positions, we modified our conclusion that,

At 125°C, the maximum stresses in both the low-k and oxide layers actually occur at -12 μm from the center of the bump, near the edge of the PI opening, with the oxide layer stress reaching approximately 340 MPa.

At -55°C, the minimum stresses occur at -9 μm from the center of the bump, rather than at the bump's center.

We apologize for any confusion caused by the previous description and have corrected the relevant section in the manuscript.

Regarding your question about why maximum stress does not occur at the bump center, the primary reason is the mismatch between the thermal expansion coefficients of the chip and substrate. This mismatch creates shear forces as the bump experiences temperature changes, resulting in tensile stress on one side of the bump and compressive stress on the other. Since both the upper layer of the bump and the BEOL layer are on the same side, the shear forces transmitted through the copper pillars in the PI opening led to the maximum stress occurring near the edge of the PI opening rather than directly at the center of the bump.

Corresponding modification is marked in blue in the text.

Comments 7: In chapter 3.5 authors check how variations in EMC thickness, mold compound CTE and polyimide opening size affect bump and BEOL stress. Interesting findings, however, I was bit surprised authors had so small PI openings comparing to bump size. It would be even more interesting to see bump and BEOL stresses when different loading of Cu layers is used in BEOL. Normally one is quite limited in design freedom here, but checking with min and max Cu densities allowed by foundry design rules would be interesting.

Response 7: Thank you for your detailed review and insightful comments regarding our findings in Chapter 3.5.

We acknowledge the oversight in the reported polyimide (PI) opening sizes, where we mistakenly labeled the radius as the diameter. We have corrected this in the revised manuscript and appreciate your understanding regarding this issue.

In response to your suggestion about evaluating the impact of different copper layer densities on bump and BEOL stresses, we conducted a parametric analysis of the copper density within the BEOL structure. We modeled six scenarios of different copper density, specifically as 30%, 40%, 50%, 60%, 70%, and 80%. The results of our simulation indicate the following maximum principal stresses in the BEOL sub-model for each respective copper density,

30%: 432.7 MPa

40%: 436.5 MPa

50%: 439.2 MPa

60%: 441 MPa

70%: 441.9 MPa

80%: 442.2 MPa

Our findings suggest that while the stress in the BEOL layer increases with higher copper densities, the overall increase is relatively modest.

Comments 8: I suggest authors to enlarge legends for figures 6, 8, 9 ,10; they are difficult to look at.

Response 8: Thank you for your constructive feedback regarding the legends in Figures 6, 8, 9, and 10. We have taken your suggestion into account and enlarged the legends for these figures in the revised manuscript to enhance their readability.

Reviewer 2 Report

Comments and Suggestions for Authors

This paper explores the comparison of FEA with testing of hybrid integration of chip packaging. They did a very systematic study considering all the important factors, including the chip BEOL as well as the levels of packaging. They appropriately separated the analysis and gave excellent understanding to the point that they made recommendations as to the points of highest reliability risk, all confirmed by experiment.

Author Response

Thank you for the reviewer's positive evaluation!

Reviewer 3 Report

Comments and Suggestions for Authors

Thank you for a good manuscript! In general, the modeling is of a classical nature and provides an answer to few number of technological and circuit engineering problems connecting with BEOl.

But I have a number of comments regarding the structure of the manuscript and its formal content:

Methodologically, you must indicate geometric limits (minimum and maximum topological norms) that can be calculated using the methodology presented in the manuscript. Qualitative judgments are sufficient here. You are indicate 28nm process technology, although similar work was carried out for 7nm technology (E. De Mesa, T. Wagner, B. Keser, J. Proschwitz and B. Waidhas, "Flip-Chip Chip Scale Package (FCCSP) Process Characterization and Reliability of Coreless Thin Package with 7nm Si Technology," 2022 IEEE 72nd Electronic Components and Technology Conference (ECTC), San Diego, CA, USA, 2022, pp. 266-270, doi: 10.1109/ECTC51906.2022.00050). Works with higher design standards are already mentioned in introduction of your manuscript.

Note on Fig. 5 - horizontal scale is not respected. Immediate transition from 30 to 100 degrees. The position of the Figure in manuscript allows the horizontal axis to be extended so that the enveloping lines between the experimental points go smoother.  Also in section 3.1. it is necessary to give a comment on why the temperature points presented for experiments were chosen. Since there are no data from 30 to 100 degrees - the most working interval for IC.

In the conclusion, it is advisable to provide references to testing methods that can confirm the assertions obtained from modeling. This would look great - you give recommendations on how to correctly compose a design, readers use one and practically confirm one with tests on the standards of which there are references (mentions) in text. All in one place!

Author Response

All of the modifications corresponding to current reviewer are in red color in the manuscript.

Comments 1: Methodologically, you must indicate geometric limits (minimum and maximum topological norms) that can be calculated using the methodology presented in the manuscript. Qualitative judgments are sufficient here. You are indicate 28nm process technology, although similar work was carried out for 7nm technology (E. De Mesa, T. Wagner, B. Keser, J. Proschwitz and B. Waidhas, "Flip-Chip Chip Scale Package (FCCSP) Process Characterization and Reliability of Coreless Thin Package with 7nm Si Technology," 2022 IEEE 72nd Electronic Components and Technology Conference (ECTC), San Diego, CA, USA, 2022, pp. 266-270, doi: 10.1109/ECTC51906.2022.00050). Works with higher design standards are already mentioned in introduction of your manuscript.

Response 1: Thank you for your thoughtful feedback and for highlighting the need for clarification regarding the geometric limits of our modeling methodology. We appreciate your suggestion to indicate the minimum and maximum topological norms that can be calculated using our approach. In our manuscript, we utilize a multilevel sub-modeling technique that enables us to accurately simulate features ranging from millimeters down to nanometers. This flexibility allows our methodology to be applicable across a wide spectrum of current and emerging technologies, including both 28nm and 7nm process technologies, or even finer technology nodes. According to your suggestion, citation of the literature of 7 nm Si technology is included in the introduction.

Comments 2: Note on Fig. 5 - horizontal scale is not respected. Immediate transition from 30 to 100 degrees. The position of the Figure in manuscript allows the horizontal axis to be extended so that the enveloping lines between the experimental points go smoother.  Also in section 3.1. it is necessary to give a comment on why the temperature points presented for experiments were chosen. Since there are no data from 30 to 100 degrees - the most working interval for IC.

Response 2: Thank you for your valuable comments. We have adjusted Figure 5 to extend the horizontal axis from 30°C to 100°C, ensuring that the scale is respected.

We have included a discussion in Section 3.1 regarding the rationale for selecting the temperature points for our experiments. Actually, the change in warpage between 30°C and 100°C is relatively minimal and primarily shows a decreasing trend. This is attributed to the fact that the material parameters within the package do not significantly change within this temperature range. In addition, the package has initial warping when cooled to room temperature after curing. Then, when the temperature increased from 30°C to 100°C, the relaxation of residual stress causes the decrease of warpage.

In contrast, temperatures above 100°C result in substantial changes in the material properties (e.g., EMC, PP, core, underfill, and solder mask) as these materials approach their glass transition temperatures. This leads to significant variations in warpage behavior, which is why our experimental design focused on observing these more pronounced changes at elevated temperatures.

Furthermore, our analysis of simulation data indicates that warpage decreases approximately linearly between 30°C and 100°C, reinforcing our choice of focusing on higher temperatures for observing significant shifts in warpage behavior.

We have referenced relevant literature that supports these findings, including studies by Xia et al. (Warpage Behavior Study and Optimization for Ultra-Thin POP Memory with Multi-Stacked Chips. In 2021 22nd International Conference on Electronic Packaging Technology (ICEPT), 2021) and Fu et al. (Chip package interaction (CPI) reliability of low-k/ULK interconnect with lead free technology.in 2010 Proceedings 60th Electronic Components and Technology Conference (ECTC), 2010), which also utilize the Shadow Moiré method to make measurements at the same temperature points.

Comments 3: In the conclusion, it is advisable to provide references to testing methods that can confirm the assertions obtained from modeling. This would look great - you give recommendations on how to correctly compose a design, readers use one and practically confirm one with tests on the standards of which there are references (mentions) in text. All in one place!

Response 3: Thank you for your insightful comments regarding the conclusion of our manuscript. We appreciate your suggestion to include references to testing methods that could support the assertions made from our modeling. In response, we have added relevant literature [27-28] on the reliability of the BEOL layer and the associated testing methods to the discussion and conclusion section. We have also included a statement emphasizing the synergy between our numerical findings and experimental testing for assessing cracking risk more effectively.